# Issues and Implications of Readiness to Change

**Yousef Ahmad Alolabi \*** , **Kartinah Ayupp and Muneer Al Dwaikat**

Faculty of Business and Economics, Universiti Malaysia Sarawak Kota Samarahan, Kuching 93000, Malaysia; akartinah@unimas.my (K.A.); maldwaikat@yahoo.com (M.A.D.)
**\*** Correspondence: y.olabi@gmail.com

**Abstract:** In light of readiness to change, organizational readiness has received little attention with the extensive assessment of individual readiness to change. (1) Background: Therefore, this conceptual paper aims to address the need for change at the organizational level through the lenses of Lewin theory, organizational change theory, and social exchange theory. It will identify issues and implications in readiness to change at the organizational level; (2) Methods: The primary method used in the study was mainly a literature review to add neglected factors driving change such as contextual factors and technology. (3) Results: The paper shows how various players and other determinants of successful change implementation can derail the organization's readiness to embrace change. (4) Conclusions: The paper adds to the available knowledge on how technology is likely to affect organizational willingness to change. The study suggests various solutions that seek to address the issues on organizational readiness to change. Hence, this study may provide organizational managers with takeaway implications on change management for policymakers and practitioners to improve an organization's preparedness towards change implementation.

**Keywords:** readiness; change; change implementation; organizational change; change management

## 1. Introduction

Business organizations are experiencing numerous changes in their settings. Managers have an essential role in meeting the variations head-on to enhance conformity to the new environments. Organizational change is a constant process that significantly affects the efficiency of the organization (Cunha-Cruz et al. 2017). Therefore, it is essential to pay attention to different change alerts from within and outside the organization. The organization is required to make steady adjustments to change more efficiently and quickly. However, the execution of change implementation requires the development of a blueprint developed more efficiently to enhance the organization's success.

Typically, change includes a wide range of activities that exist in the company. The common changes in an organization include layoffs or downsizing, restructuring operations, and reorganizing teams. Some organizations engage in practices such as mergers, reengineering, and the development of new technology to ensure their readiness for organizational change (Weiner et al. 2008). Changes are meant to reorient and reorganize how the organization conducts its activities. The main goal of change is to identify new and improved methods to ensure optimized use of available resources and the general capabilities to ensure the organization's increased ability in value creation and provide enhanced returns to stakeholders.

The incorporation of new processes helps everyone in the organization conduct their jobs better and increase the positive contribution to the organization's current needs. An organization that fails to embrace change is likely to lose its competitive edge and fail to ensure conformity to the needs and demands of customers and stakeholders (Bank et al. 2017). Factors that determine the extent of change in an organization exist internally and externally. The economy is likely to pose a significant impact on the success of the organization (Vakola 2014). Sometimes change comes from adopting a new technology whose aim is to

increase productivity and communication that form part of the organization's readiness to change.

The use of updated technology helps in the exploration of new markets to get better opportunities. They can enhance, modify, and create new products that will keep a loyal customer base. In some cases, it is understood that changes arise within the organization due to political pressures, identity pressures, and growth pressures (Billsten et al. 2018). For instance, some organizations adopt change programs due to a new vision from a new Chief Executive Officer (CEO). Therefore, the new concepts and ideas might be incorporated to suit the needs of the new leadership. This results in new and innovative concepts in the organization. Change is known to have emotional and physical effects on people; hence, it is essential to show how change is likely to improve the workforce environment.

However, resistance to change is a significant setback to an organization's readiness to implement new ideas. Change brings uncertainty, specific attachments, a perceived breach of the psychological contract, and negative perceptions. Sometimes past experiences with change also affect the willingness of organization members to accept change, especially if it went wrong. Therefore, managers are responsible for managing resistance situations by integrating various approaches such as education and communication, negotiation and agreement, participation and involvement, and explicit and implicit coercion (Weiner et al. 2009). At the organization level, readiness to change is affected by organizational culture, contextual factors, leadership behaviour, and technological impact.

The paper deals with Issues and Implications of Readiness to Change; therefore, to understand this, the literature review is organized with research on Organizational Readiness to Change, Need for Change, Expected Outcomes from Readiness to Change, Relationship between Determinants and Outcome of Organizational Readiness for Change, Different variables impact on the readiness of an organization, the framework model for the measurement of readiness to change, Contextual Factors, Organizational culture, leadership, and technology and their correlation with one another.

## 2. Methodology on the Overview of the Literature Review

The literature review deals with different concepts about organization readiness. It starts from the background concepts, then deals with the new concepts, and ends with the applied concepts. Organization readiness to change is crucial for the members to have a shared belief in the efficacy of change. The readiness of change in an organization depends on the preparedness of various players in the industry as well. The need for change is a new concept that deciphers the very concept behind organization readiness to change, and the applied concepts are related to the influence of organizational culture on organization readiness and the outcomes associated with it.

## 3. Organizational Readiness to Change

Readiness to change in an organization is considered a multi-level and multi-faceted construct. It can be present at the individual, group, department, unit, or organizational level. At the organizational level, readiness for change is defined as the shared resolution by organizational members to implement change (Al-Maamari et al. 2018). It is also crucial for the members to have a shared belief in the efficacy of change. The readiness of change in an organization depends on the preparedness of various players in the industry. People are the most significant concern in assessing the issues for change readiness. Technology is another factor that determines whether the organization can effectively undertake innovative changes to keep up with the fast-changing organizational activities. Lewin (1947) states that theoretical formulation is needed to revive the social sciences. According to him, three goals are dominant, the integration of the social sciences, the transition to concern about "dynamic problems of group life changing from social bodies", and the development of "new tools and techniques for social research". None of the activities can be explained in their own terms, only as part of the operation of different processes that fluctuate as a function of fundamental forces and tensions. Cummings et al. (2016) explored how and

why it came to be understood as the foundation of the new subfield of change management, how it has influenced change theory and practice to date, and how questioning this assumed foundation can foster innovation. For a change, it is necessary to create added value by using the existing ones.

Employees are responsible for the implementation of change. Therefore, it is critical to assess whether they are ready to implement effective organizational changes. The varying levels of readiness to change in different organizations depend on how the members of the organization value change and the likely implications it will have on their work environment (Von Treuer et al. 2018). They are expected to appraise the significant determinants of change implementation capability, such as resource availability, task demands, and situational factors. High organizational readiness is characterized by the willingness of organizational members to initiate the programs of change through greater efforts and cooperative behaviour.

## 4. Need for Change

The motivation theory explains why commitment to change is considered a function of change valence. Members of the organization play a vital role in the success of the organization's readiness for change. Several conditions determine their level of commitment towards embracing change. For instance, it is crucial to consider whether they understand the value of change and various benefits that are likely to be accrued from the entire process (Weiner 2009). Increased numbers of organizational members that value change will result in a willingness to participate in the implementation of change. The issues with changing readiness arise from the disparate drivers of change management.

In some cases, members of the organization value change because they understand the contribution and the urgency of change outcomes. They might display a high level of cooperation given the positive impact of the change process in sorting organizational issues. To some extent, the preparedness and readiness for change are boosted because members value the personal benefits likely to be derived from evolution (Weiner et al. 2009). Therefore, change valence that results from a different disparate reason is a potent determinant of the levels of commitment to change. For readiness to change, the critical concern is whether members of the organization value the whole idea of change implementation to express their devotion.

## 5. Expected Outcomes from Readiness to Change

The greater level of an organization's readiness to change implies that change implementation is likely to be successful. High levels of change readiness are an indication that an organization's stakeholders have the willingness to initiate change through instituting various procedures, policies, and practices (Weiner 2009). They tend to exert increased effort to support the change programs and portray significant levels of persistence to overcome multiple setbacks and obstacles in the implementation stage. The members are highly motivated, as evident from the prosocial and other behaviours with a close relationship with change implementation. Therefore, members are likely to come up with various measures that target exceeding the job requirements. Successful implementation of the change program is the proximal outcome of the organization's readiness for change. Effective implementation involves the quality and consistency of adopting and using the new idea, program, process, or technology at the easy stages. When the organization's readiness for change is low, its members may be reluctant and less perseverant in implementing the change programs (Desplaces 2005). However, it should be noted that readiness for change is not a guarantee for the success of a complex change program or process. Various factors that determine change success include safety, quality, efficiency levels, and anticipated outcomes Sometimes, organizational members can also offer wrong judgments regarding the level of the organization's readiness to change. Such mistakes arise from underestimating or overestimating the required collective capability for change implementation. Therefore, efficacy judgments should be based on direct experiences and

rich and accurate information to ensure that it is more predictive than judgments made based on erroneous and incomplete information.

## 6. Relationship between Determinants and Outcome of Organizational Readiness for Change

Generating organizational readiness to change is a difficult task. The social cognitive theory and motivational theory provide several conditions and circumstances that dictate successful change implementation. Figure 1 outlines the steps that lead to change implementation.

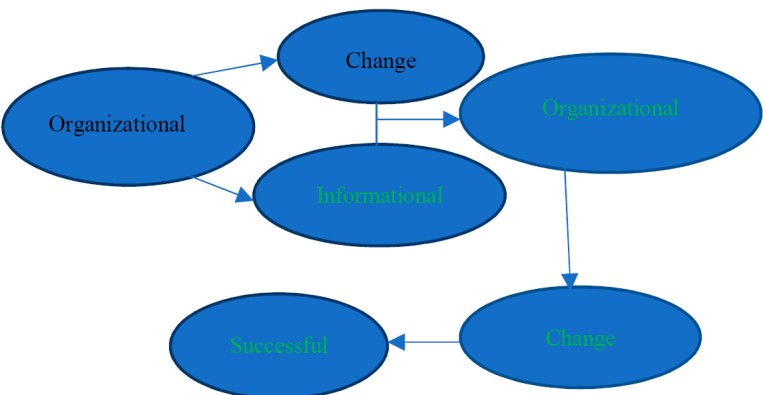

**Figure 1.** The relationship between the determinants and outcomes of organizational change.

## 7. Conceptual Framework

Different variables impact the readiness of an organization towards accepting change programs. The framework model for the measurement of readiness to change is divided into four perspectives. The use of the framework helps provide a research model that integrates various issues that influence the readiness to change from the organizational perspective (Vaishnavi et al. 2019). The variables are categorized into independent and dependent variables, as shown in Figure 2. From left to right, independent variables are the factors that include contextual factors, leadership, organizational culture, organizational capability, and technology, and organizational readiness to change is the dependent variable.

## Independent variables     Dependent Variable

**Figure 2.** The Conceptual Framework of organizational readiness to change.

### 8. Contextual Factors

Apart from issues arising from the organizational members such as leaders and employees, experts and scholars in the field of change management argue that various conditions still exist that affect the process of change. One of the issues is the organizational culture that encompasses its levels of innovation, learning, and risk-taking to ensure that the organization gets ready for change implementation (Vaishnavi and Suresh 2020). The other factors concern the organization's flexibility in policies and procedures towards a favourable climate change (Benzer et al. 2017). An example of a positive organizational environment is good working relationships. Past experiences with change programs also affect the level of an organization's readiness to change. For instance, an organization that has experienced positive outcomes from the change process is likely to embrace readiness. This is because the change implementation aligns with the organization's values, considering that it affects the member's valence to change.

### 9. Organizational Leadership

Leadership involves persuasive practices aimed at ensuring that members of the organization comprehend and reach a decision on various organizational undertakings by determining activities that should be done, methods involved, and multiple means to enable individual and collective efforts to achieve the shared objectives (Al-Hussami et al. 2018). Change in an organization involves specific actions that demand the leadership role to change the existing procedures. The transformational style of leadership is considered an

essential tool for enhancing an organization's readiness for change. It entails preferred organizational transformations seeking to improve the performance of the organization. Change management is an objective of the organization. Thus, management functions such as organizing, controlling, leading, and planning form part of the organization's readiness for change. Leaders are expected to develop innovative ideas that involve a clear vision and long-term thinking to ensure that the organization is prepared to execute change management programs (Asbari et al. 2021). Leaders must understand that there is a need for organizational readiness to change, considering the positive impacts that arise from it. For instance, it seeks to improve the efficiency and effectiveness of the changes through procedures and policies seeking to introduce new technologies and redesign an organization's entire business process.

The types of leadership in an organization determine the level of resistance in implementing change programs in an organization. High resistance arises from within the members, such as organizational employees hence becoming a significant issue in readiness to change. Change has a direct link to the management of people and the role played by leaders in influencing behaviour. Leadership is characterized by the capability to control the actions of other organization members who contribute to the organization's readiness to change (Asbari et al. 2021). Lack of influence is also accompanied by resistance; hence, the organization faces a setback in its readiness to change. As mentioned earlier, transformational leadership is critical in readiness for change by acting as good role models and motivating employees' actions. They also have a close relationship with employees, thus influencing large groups of people to accept change.

The other aspect of transformational leaders is compelling communication that forms an essential part of preparedness for change implementation. Transformational leaders are characterized by effective interpersonal skills that are fundamental in the performance of change in an organization (Guerrero and Kim 2013). Some leaders have charismatic traits that determine the extent to which they influence, mentor, and encourage their followers by inspiring them regarding the need for personal development and organizational development (Miake-Lye et al. 2020). They are assumed to be role models to their fellow organizational members and thus encourage them in the challenging business environment to embrace change programs by taking more initiative.

Organizational readiness to change should be reviewed from time to time before deciding on making organizational changes. Leadership in an organization has a responsibility towards the efficacy of change implementation. Non-committed leaders are a setback to the readiness of an organization to embrace change. A leader serves as an important figure responsible for the coordination of various organizational practices. A practical leader moderates the organization's activities and enhances proper coordination of activities to embrace change readiness. They act as both agents and serve as consultants, researchers, and trainers. The consultancy role entails performing various efforts to ensure that organizational members have exposure to the organization's external data and management of internal data. Leaders are also responsible for helping the organizational members learn about collecting, processing, and usability of data to solve problems.

As researchers, leaders help members effectively evaluate the validity of information and other action plans implemented in the organization. The three roles performed by leaders in promoting organizational readiness for change include informational, interpersonal, and decisional, as presented by (Guerrero and Kim 2013). The interpersonal role involves creating an interpersonal relationship that entails a figurehead, liaison, and leader. The informational role of a leader requires control of information in the organization through acting as spokespersons, disseminators, and monitors of the full readiness of the organization to change. Leaders ' decision-making roles encompass taking responsibility in decision making, innovative practices, handling disturbances, negotiation roles, and resource allocators (Roos and Nilsson 2020). Therefore, it is clear that a close relationship exists between the leader's responsibility and the overall success in the readiness for change implementation in an organization. Leaders have a significant influence on the

organizational readiness for change through their ability to execute their roles effectively. Leadership failure to perform functions effectively is likely to be a significant issue in different ways.

## 10. Leadership Issues in Readiness for Change

Poor leadership practices result in various issues in readiness for change. The problems range from lack of employee involvement, wrong communication strategies, and organizational complexity. The standard issue that results from poor leadership is a lack of employee involvement. Employees tend to fear change. Therefore, it is the responsibility of leadership to involve employees in the change process and keep them updated on various developments within and outside the organization (Roos and Nilsson 2020). This is the biggest mistake committed by several organizations. As a result, members of the organization experience fear and increased laxity in embracing the new culture. The initiative's success depends on the leadership commitment towards ensuring that they are involved in the process as much as possible.

Leadership is required to listen to their views and opinions, which accounts for their output and assures all members that the change process is for the interest of everyone and it will bring positive outcomes. Failure by leadership to provide the required and sufficient resources to organizational members is a setback to the new change development in the industry (Riley et al. 2021). It is crucial to stay connected with the employees considering the challenges experienced in explaining the organization's vision. The readiness to change is possible when change management initiatives are aligned with the capabilities of individuals in the organization.

The other issue of concern from leadership is the lack of an effective communication strategy. Some leaders suffer from the inability to provide practical strategies for communication. It is common in various organizations to assume that announcement of plans of the organization to undertake a particular change program guarantees that the organizational members will automatically adjust with the new change developments (Kelly et al. 2017). It is considered ineffective to introduce change in an organization, thus countered by a high resistance level. Leaders must identify strategies to communicate the new ideas, programs, and procedures and convince members of the organization of the overall value likely to be accrued from the change program. Once the employees understand the consequences of change on their work environment, they will be ready to embrace and adapt to the new change programs.

Some organizational changes are highly complex; thus, leaders are responsible for enhancing organizational readiness towards countering the challenges likely to arise. The everyday complexities that members of the organization fail to understand include complex products, systems, and processes that contribute to barriers to the readiness of the organization to change (Sanders et al. 2017). It is the responsibility of leaders to ensure that everyone understands the complex processes. Some leaders fail to adopt a skillful and keen approach that tackles the fast-growing complexity. Leaders are expected to adopt some of the practices that include employing quality, diligent, and effective change management approaches (Mangundjaya and Gandakusuma 2013). Leaders are also required to exhibit a high level of transparency and accountability throughout getting the organization ready for change. Accountability helps to foster commitment and desire to fix various problems to yield the required results. Ineffective leaders pose a challenge to the processes, management, culture, and employees and fail the optimal functioning of the organization (Lehman et al. 2002). Some leaders fail to admit various gaps leading to misalignments in the organization without addressing the shortcomings.

## 11. Organizational Culture

The perceptions of employees on the existing organization culture in human open system values and human values would have a close association with increased levels of change readiness, predicting the level of change in an organization. Various factors

determine failure for change readiness in an organization, and some are critical considering that the attitude of employees plays a significant role in the change program (Madsen et al. 2006). Resistance towards change implementation and outright failures in attempts to introduce change can be traced by the inability of the organization to create a positive culture for change, which is fundamental in getting organizational stakeholders ready for change (Douglas et al. 2017). Many organizations assume that change will be successful and therefore tend to bulldoze the implementation of change before changing the psychological readiness of individuals and groups in the organization. Consequently, it is evident that the organizational culture fosters an environment that influences change readiness among employees. The extent to which employees hold positive views about the importance and the need for organizational change is quite helpful in determining change acceptance. Positive employees believe that change will have a favourable implication for themselves and the organization in general (Nordin 2011). Some approaches also focus on determining employees' perceptions towards the organization's preparedness towards change programs in large-scale initiatives. The organizational culture has the effect of reshaping capabilities and employee readiness to impede organizational changes. Clearly, the extent to which organization members' perception of change helps predict whether the organization achieves positive change outcomes.

Organizational culture has a three-dimensional view consisting of (i) values, (ii) assumptions, and (iii) artifacts. Assumptions involve certain beliefs about the nature of human beings and the overall organizational environment residing deep below the surface (Madsen et al. 2006). Values are defined as shared attributes and beliefs and specific rules that help govern employees' behaviour and attitudes in an organization (Nordin 2011). They also entail attempts to make social and personal modes of conduct that are acceptable relative to others (Dhingra and Punia 2016). On the other hand, artifacts are defined as more visible behaviours, language, and material symbols available in an organization. Values play a central role in gaining an understanding of the organizational culture. Equally, it is noted that the organization's culture is focused on providing specific values.

An organization is expected to develop competing values by creating a clear framework of the overall organizational culture. The framework consists of internal and external parameters like human relations, open systems, rational goals, and internal processes. All these parameters can be scaled from flexibility to control as shown in Figure 3. The framework is essential in exploring competing demands within the organization in two dimensions. Therefore, the classification of an organization is done based on whether it values control or flexibility in structuring the organization. Different organizations have different opinions regarding whether they need to adopt an inward focus towards the internal dynamics or develop an external dynamic focus towards their change environment (Ritchie and Straus 2019). Organizational culture is characterized by four quadrants commonly referred to as culture types. Each quadrant in the framework has specific characteristics. For instance, an organizational culture whose focus is emphasizing the need for strong human relations and values develops a goal to foster high levels of organizational morale and cohesion among the members through programs such as training and development, participative decision making, and open communication (Jones et al. 2005). Another quadrant in the framework pertains to available systems orientation that values the need for having high employee morale and strong emphasis on innovation and development. The goals can be achieved by fostering readiness and adaptability, adaptable decision-making, and proactive communication.

The high internal process values dimension is where the organization is focused on promoting control and stability through precise communication, proper information management, and data-based decision-making. The last quadrant in the framework is the role of the organizational culture in addressing rational goal orientation. An organization with a reasonable goal orientation promotes increased efficiency and productivity. This is possible through planning and goal-setting practices, centralized decision-making, and instructional communication. The centralized decision-making and instructional types of

culture are characterized by lower morale and cohesion among the employees (Grimolizzi-Jensen 2018). The four organizational cultures are understood to be mutually exclusive and exist in a single organization with specific values likely to have a high dominance over others.

**Figure 3.** The competing values framework (CVF). Source: A theory of organizational readiness for change by Jones et al. (2005).

## 12. Organizational Culture on Readiness to Change

An organization can experience different perceptions in its readiness to change. This is because various individuals exhibit specific differences with the culture of the organization polarizing the members' attitudes, beliefs, and intentions. Flexible organizational culture coupled with solid structures and supportive change climates is likely to provide a conducive environment that impacts successful organizational readiness to change (Snyder-Halpern 2001). Organizations that employ mechanized strategies are likely to experience control and inflexibility. Employees who agree and perceive the workplace to dominate both open system and human relations values are highly likely to hold positive views and beliefs about the organizational change process. Orientation of human relations is characterized by the tendency to engage in human resource training and development to ensure that the capability and confidence of individuals in the organization are highly boosted in enhancing their preparedness towards undertaking challenges and issues that are likely to emerge from the change process (Levesque et al. 2001). The open systems provide dynamic and innovative practices. They suggest that employees with the perception that organizational culture is an available system are likely to be equipped with positive attitudes that are significant in the entire organizational change process. The factors of human involvement and communication form part of the characteristics of human relations.

## 13. Organizational Technology and Readiness to Change

Readiness to change in an organization also faces issues in line with the level of technology available in an organization. The level of development in technology and innovation depends on certain strategic organizational practices. Every organization seeks to develop successful technology management by providing an innovative strategy that determines its extent in readiness for change (Rafferty and Minbashian 2019). Various uncertainties and risks face changes that take place in an organization, hence demanding organizational agility. It is crucial to balance agility and technology and innovation man-

agement to ensure that short-term efficiency and long-term efficiency achieve the required effectiveness in change management.

An organization is expected to have solid dynamic capabilities to address particular organizational challenges in the vigorous competition of innovation practices. To balance the conflicting demands of agility in the dynamic environment, the organization should consider various factors. First, the design systems and processes are an essential consideration seeking to assess, identify, and develop technology based on opportunities presented in the organization (Kwahk and Lee 2008). The level of preparedness also relies on the need to ensure protection against emerging technologies that can affect change implementation relative to competitors.

Communication needs and organizational efficiency should be aligned to turn data into information for better decision-making. For instance, the rising interest in big data is a significant consideration on whether the organization has effectively analyzed the available before deciding whether it is possible to embrace change implementation that aligns with the organization's goals (Kwahk and Lee 2008). Therefore, it is essential to determine whether computer technology is highly developed and efficient, as well as effective in use. The development of employees through training and development also seeks to ensure that they possess the necessary skills in the dynamic technological environment before the change is implemented.

Technology and innovation demand the involvement of all organizational levels and making necessary efforts to enhance the required skills. It is crucial to determine how dynamic the environment is to ensure the necessity and emphasis on skill enhancement in achievement at the individual and firm levels (Ritchie and Straus 2019). Therefore, processes of change management rely on organization technology by assisting in introducing innovations in the organization. Non-innovative organizations, due to inadequate levels of technological development, are likely to face critical challenges, thus affecting the entire organizational change readiness (Ritchie and Straus 2019). Technical preparedness in change management involves engaging partnerships and purchase of technology requirements. The standard methods of acquiring technology include buying and collaboration, which entail acquisitions and mergers, joint ventures, contractual agreements, and other forms of technology from third-party providers and other external sources.

However, technology can be acquired internally by engaging in research and development of new systems and products. It also involves developing new processes and reconfiguring the organizational way of doing things (Katsaros et al. 2020). It entails the structure of the organization and redesigning an assembly line. For instance, an organization can develop robotics added in the manufacturing process due to internal drivers to ensure the firm purchases robots to acquire capability in adding robotics to the entire assembly process (Katsaros et al. 2020). The creation of technology can also involve entailing the new technologies and innovations resulting from exploiting space in the environment through new techniques in business development and entrepreneurial activities. Owing to the role of technology in change management, any issue related to technology is likely to derail change implementation through the organization's readiness to change.

For instance, readiness can be determined by the rate of development in technology in the current organization. Technology is fast-changing as compared to the capacity of people towards learning and keeping up with the demands. For instance, the emergence of machine learning and artificial intelligence aims to provide an opportunity seeking to capitalize on the needs of employees through behaviour learning and anticipation of different actions likely to take place and thus affect the change implementation activities ( Matthysen and Harris 2018). Besides creating a simplified workflow in managing change, technology is also significant in ensuring that the workforce stays informed concerning the relevance in real-time about change information.

Technology can necessitate the organization's senior leaders to effectively communicate information, news, and changes likely to impact team members. It also plays a crucial role in measuring the success of various change initiatives throughout the lifecycle

of change implementation. It helps people to understand the impact of change, the level of commitment to required actions, and ensuring that the right technology is put in place (Matthysen and Harris 2018). Therefore, the role of technology in the process of change management cannot be underestimated. This is because people, content, and silos of information are considered pervasive across various businesses. Organizations are demanding more excellent connectivity across different organizational resources to enhance improved clarity, agility, and focus on improving quick response to changes accrued from digitization. The complexity of technological systems is likely to affect the process of change implementation.

Some enterprises are also operating in monolithic legacy technologies saddled with complexity in providing designated operations for specialized needs, thus lacking interoperability. The level of employee productivity, which is also a key consideration in organizational readiness for change, depends on the organization's ability to enhance the ability to leverage change implementation. Successful change management is essential in threading various information, people, and knowledge to form a single point in productivity. The use of real-time tools in collaboration such as video calling, live chat, and voice provides a killer combination that a leg up on levels of competition aimed at driving a high level of success in the new world of work.

## 14. Organizational Capability and Readiness to Change

The idea of business capability has its foundation in competitive advantage ground. The concept of competitive advantage is viewed in a resource-based dimension with a business having heterogeneous capabilities and resources. This means that the business's competitive performance and set strategies significantly depend on the organization's specific capabilities and resources. Organization capabilities in terms of intellectual properties and technology should always be associated with the managerial and organizational processes as they are crucial in sustaining the organization's performance (Hindasah and Nuryakin 2020). For instance, higher capability levels are linked with a sustained process irrespective of whether it is on product development, employee satisfaction, or financial performance.

To enhance readiness for change, organizations are working towards demonstrating and providing timely responsiveness to effective and efficient coordination and deployment of internal and external competencies (Weiner 2009). This is both in the future and current global market economy and having organization flexible in manipulating both existing and new ideas without forgetting the dynamic capabilities required when responding to various shifts and trends in both external and internal environment and capabilities are necessary for change (Huang and Li 2017). Some of these capabilities should be reshaping capabilities, and potential capabilities as potential capabilities will take charge of sustaining an organization's daily performance as they are not supportive to help businesses management and respond to trends effectively. Potential capabilities are crucial as they enable the management of the current organization's conditions, thereby facilitating the smooth implementation of the other changes and trends.

On the other hand, the reshaping of capabilities is also significant as it will enable organization taskforces to have a sense of commitment and motivation and work towards achieving the set organization objectives and goals. This involves developing all organization systems and resources required to complete and keep a business's future route. The management of these capabilities that will keep driving the organization's performance effectively and consistently should be proactive for an organization to achieve the intended change (Hindasah and Nuryakin 2020). Therefore, the development and strength of both reshaping and potential organizational capabilities lead to successful change implementation. However, effective change yield can be undermined when a business has reshaping capabilities of low levels, as there is a strong bond between the execution of successful organization changes and reshaping capabilities (Ruest et al. 2019). Therefore, reshaping

capabilities should be highly inclusive whenever a change is expected as their impact on the organization's current performance is feeble.

It is vital to understand the link between readiness for organizational change and reshaping capabilities in a transition process. This factor determines the organization's readiness level for change. This is determined by evaluating the organization's workforce attitude toward different change events and examining the organization's effective change management capability (Øygarden and Mikkelsen 2020). On the other hand, readiness for change should entail employee willingness and motivation, as reshaping capability also entails the organization's skills, ability, and knowledge to successfully perform all the necessary operations required to implement change successfully (Øygarden and Mikkelsen 2020). There should be effective communication and involvement throughout developing employees in all divisions to have readiness for change. This is a vital change implementation strategy that is always inclusive as employee change perception of overall readiness in every organization division needs to be evaluated to check their readiness before implementing any change event.

## 15. The Solutions in Organizational Readiness to Change

The issues experienced in readiness to change can be solved by taking necessary actions at an organizational level. For instance, employees may not agree with change based on their level of awareness about its impact on the work environment (Qiao et al. 2021). Therefore, leaders should provide training opportunities required in performing certain tasks. Leadership and employees need to collaborate for change to take place. Some of the strategies include building a devoted leadership team, working with vision, controlling staff turnover, and assessing the organization at every stage (Qiao et al. 2021). A positive organizational culture that embraces change and involves effective past experiences in leadership implementation is also key in ensuring change readiness.

The organization should be aggressive in the introduction of new technologies that are fundamental in change management. However, technology can be coupled with challenges that demand immediate actions, such as focusing on employee training on how to use the technology. The organization should also consider the benefits of technology relative to the complexity and costs of acquiring new technology. More investment in innovation and creativity is also crucial in addressing technical issues to change.

## 16. Conclusions

Organizational readiness is a shared psychological state involving the organization members getting involved and expressing their commitment to implementing organizational changes. However, various factors are considered significant concerns towards the change readiness. Adoption of technology is regarded as a critical area towards increasing the effectiveness of organizational activities. Organizational culture involves employees' perceptions of the existing organization practices in open system values and human values and how it would closely associate with increased readiness levels, predicting the level of change in an organization. A positive organizational culture and a welcoming workforce attitude towards difference are likely to affect change readiness positively.

## 17. Academic and Practical Recommendations

From the conclusion, it is evident that organizational readiness can improve the coordination and synchronization among an organization's body to implement a change or process. This paper deals with the correlation between organizational readiness and the factors affecting it. However, there is a need to understand how these factors actually enhance organizational readiness from a psychological perspective. Further research can be done on the training that actually changes organizational readiness for the benefit of the organization. In the practical domain, there is a need for experiments on different ethnical samples to study how organizational readiness varies across different races, colours, and creeds.

**Author Contributions:** Conceptualization, K.A. & Y.A.A.: Methodology, Y.A.A. & K.A. Validation, Y.A.A. and M.A.D.: Formal anysis, Y.A.A.; Investigation, Y.A.A. and M.A.D.; Resources, Y.A.A.; Data tabulation, Y.A.A. and M.A.D.; Writing—draft preparation, Y.A.A.; Writing—review, K.A. All authors have read and agreed to the published version of the manuscript.

**Funding:** The authors received no financial support for this article's research, authorship, and/or publication.

**Data Availability Statement:** Not available.

**Acknowledgments:** The authors have not received any financial support from a university or any other institution/organization. The authors are grateful to the journal's anonymous reviewers for their extremely helpful suggestions to improve the quality of the manuscript. They thank the University of Malaysia Sarawak, UNIMAS/Malaysia for their support while conducting the research.

**Conflicts of Interest:** The authors declare no conflict of interest.

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
