# Peer review of "Issues and Implications of Readiness to Change"

_admsci, doi:10.3390/admsci11040140_

Round 1

Reviewer 1 Report

An extremely interesting paper, it fits into the research gap that is the impact of technology on readiness for change. Recommended for both policy makers and managers.

Author Response

Dear Sir/Madam,

I am really grateful for the positive feedback and the encouraging review I have received. I am attaching an enhanced version after editing.

Many thanks again 

Yousef  

Reviewer 2 Report

I believe that the topic of this paper is interesting and provides an original contribution to the relevant literature

The title of the paper fits to the content

The Abstract section has a clear roadmap of the study

The conceptual framework is well established

The references section can be developed, I believe there are much more sources regarding this research topic

The paper can be proofread

The conclusion section can be reconsidered again, and it will be more functional to have a comprehensive conclusion

Policy recommendations can be provided

The academic and practical recommendations can be presented for future studies.

Author Response

Dear Sir/Madam,

Many thanks for your constructive review and comments on my paper. Kindly find attached the amended paper as per your recommendations. I made sure that the sources are enhanced (highlighted in green). I have proofread it and enhanced the conclusion section as per your recommendations. These changes are all colored in green for ease of tracking. 

I am really grateful for the time you have put to read my paper.

Many thanks in advance 

Yousef

Reviewer 3 Report

The paper topic is pertinent and current for understanding the organizational change process. Currently, change represents in organizations a strategic factor for the development of resilience.

The abstract has all the essential parts for understanding the investigation.

The authors should describe the methodology on the overview of the literature review.

It is advisable that the text has another structure, such as a section with the introduction and another with the literature review. At the end of the paper, the authors should indicate the contributions of research to the theoretical and practical framework and future investigations.

Check that the citation method is in accordance with the writing standards of the journal and correct some graphic inaccuracies, such as Figure 2.

Author Response

Dear Sir/Madam,

I hope that my message finds you always well. I am so grateful for the constructive feedback and review. Kindly find attached the amended paper with the section in green color for your kind review.

Many thanks in advance 

Yousef